# High-Frequency Irreversible Electroporation (H-FIRE) Induced Blood–Brain Barrier Disruption Is Mediated by Cytoskeletal Remodeling and Changes in Tight Junction Protein Regulation

**DOI:** 10.3390/biomedicines10061384

**Published:** 2022-06-11

**Authors:** Brittanie R. Partridge, Yukitaka Kani, Melvin F. Lorenzo, Sabrina N. Campelo, Irving C. Allen, Jonathan Hinckley, Fang-Chi Hsu, Scott S. Verbridge, John L. Robertson, Rafael V. Davalos, John H. Rossmeisl

**Affiliations:** 1Department of Small Animal Clinical Sciences, Virginia Tech, Blacksburg, VA 24061, USA; brittanp@vt.edu (B.R.P.); yukitaka@vt.edu (Y.K.); hinckley@vt.edu (J.H.); 2Department of Biomedical Engineering and Mechanics, Virginia Tech, Blacksburg, VA 24061, USA; mflorenz@vt.edu (M.F.L.); scampelo@vt.edu (S.N.C.); 3Department of Biomedical Sciences and Pathobiology, Virginia Tech, Blacksburg, VA 24061, USA; icallen@vt.edu (I.C.A.); sverb@vt.edu (S.S.V.); drbob@vt.edu (J.L.R.); davalos@vt.edu (R.V.D.); 4Center of Engineered Health, Virginia Tech, Blacksburg, VA 24061, USA; 5Department of Biostatistics and Data Sciences, Division of Public Health Sciences, Wake Forest University School of Medicine, Winston-Salem, NC 27101, USA; fhsu@wakehealth.edu

**Keywords:** high-frequency irreversible electroporation (H-FIRE), blood–brain barrier, glioma, intracranial drug delivery

## Abstract

Glioblastoma is the deadliest malignant brain tumor. Its location behind the blood–brain barrier (BBB) presents a therapeutic challenge by preventing effective delivery of most chemotherapeutics. H-FIRE is a novel tumor ablation method that transiently disrupts the BBB through currently unknown mechanisms. We hypothesized that H-FIRE mediated BBB disruption (BBBD) occurs via cytoskeletal remodeling and alterations in tight junction (TJ) protein regulation. Intracranial H-FIRE was delivered to Fischer rats prior to sacrifice at 1-, 24-, 48-, 72-, and 96 h post-treatment. Cytoskeletal proteins and native and ubiquitinated TJ proteins (TJP) were evaluated using immunoprecipitation, Western blotting, and gene-expression arrays on treated and sham control brain lysates. Cytoskeletal and TJ protein expression were further evaluated with immunofluorescent microscopy. A decrease in the F/G-actin ratio, decreased TJP concentrations, and increased ubiquitination of TJP were observed 1–48 h post-H-FIRE compared to sham controls. By 72–96 h, cytoskeletal and TJP expression recovered to pretreatment levels, temporally corresponding with increased claudin-5 and zonula occludens-1 gene expression. Ingenuity pathway analysis revealed significant dysregulation of claudin genes, centered around claudin-6 in H-FIRE treated rats. In conclusion, H-FIRE is capable of permeating the BBB in a spatiotemporal manner via cytoskeletal-mediated TJP modulation. This minimally invasive technology presents with applications for localized and long-lived enhanced intracranial drug delivery.

## 1. Introduction

The blood–brain barrier (BBB) serves a critical role in maintaining brain homeostasis by regulating transport of molecules across the microvasculature within the central nervous system via paracellular and transcellular pathways [1]. Paracellular transport of substances into the brain is primarily restricted by the presence of tight junctions between adjacent specialized brain capillary endothelial cells (BCECs). Tight junctions primarily consist of a network of transmembrane (claudins, occludin and junctional adhesion molecule), cytoplasmic (zonula occludens) and cytoskeletal (actin) proteins, which play a critical role in modulating the function of tight junction complexes [2]. Tight junctions, coupled with molecular efflux transporters present on the surface of BCECs, limits transport across an intact BBB to small, lipid soluble molecules with a molecular mass under 400–500 Da [3,4]. Though effective at isolating the central nervous system (CNS) from circulating toxins, an intact BBB also excludes most therapeutic drugs and large molecules, impeding treatment of intracranial malignancies and CNS disorders [5,6,7]. Overcoming this obstacle by inducing temporary blood–brain barrier disruption (BBBD) will be critical to the advancement of potential therapies for a wide range of intracranial disorders, including but not limited to CNS malignancies. Several therapeutic techniques capable of transiently permeating the BBB exist with the potential to improve delivery of therapeutic agents to the brain, including laser interstitial thermotherapy (LITT), hyperosmotic therapy, focused ultrasound (FUS) and irreversible electroporation (IRE) [8]. These technologies hold promise for improving the long-term outcome of patients with CNS malignancies by facilitating treatment of infiltrative tumor cells, a major source of tumor recurrence and subsequent death.

Irreversible electroporation (IRE) is a well-established, non-thermal tumor ablation method that uses short (~10–100 us) pulsed electric fields (PEFs) delivered through minimally invasive electrodes placed within the target tissue. The primary cellular effects of IRE pulses are dependent on the strength of the electric field applied to the tissue. For a given parameter set, lower electric field strengths (≤~500 V/cm) result in transient and reversible cell membrane permeabilization, which allows for targeted delivery of molecular agents and chemotherapeutics. Higher electric field strengths (≥~500 V/cm) result in irreversible cell membrane permeabilization and subsequent cell death characterized by a well-demarcated zone of ablation with minimal-to-no evidence of thermal necrosis [9]. Modification of pulsing parameters and electrode configurations allows for clinically relevant tissue ablation without collateral damage to adjacent critical structures, such as blood vessels and ducts [9]. Furthermore, when higher electric field strengths are applied within the brain, a single treatment results in a central zone of non-thermal ablation surrounded by a region of voltage-dependent reversible electroporation and transient BBB disruption (BBBD) [10]. When applied to primary CNS malignancies, this region coincides with a penumbra of infiltrating cancer cells at the tumor margin, which are otherwise protected by an intact BBB. Current therapies are limited in their ability to target and kill these infiltrating cells, resulting in high rates of tumor recurrence and short survival times, despite aggressive treatment. Thus, IRE is an attractive treatment option for management of primary CNS malignancies given its ability to permeate the BBB for targeted drug delivery to infiltrative cells extending beyond the tumor margin. Likewise, given the volume of reversible electroporation present beyond the ablation zone, IRE may be exploited as a form of electrochemoablation to further target infiltrative cells.

The safety and feasibility of IRE has been demonstrated for a number of different cancers in both animal cancer models and human cancer patients [11,12]. IRE has been demonstrated to be a technique capable of tumor ablation and BBBD in spontaneous canine malignant glioma models, whereby clinically relevant ablations were achieved with minimal adverse effects attributed to therapy [13,14,15,16]. Limitations of IRE include treatment-induced cardiac asynchrony and muscle tetany, necessitating cardiac synchronization and neuroparalytics during treatment delivery. Additionally, tissues with heterogeneous electrical properties may distort the electric field distribution induced by IRE delivery and subsequently diminish treatment efficacy.

To overcome these limitations, our team developed high-frequency irreversible electroporation (H-FIRE), the next generation of IRE technology. H-FIRE uses ultrashort (~0.5–10 us) bursts of bipolar pulsed electric fields to non-thermally ablate tumors with minimal muscle contraction and cardiac asynchrony, negating the need for neuroparalytics and cardiac synchronization during treatment delivery [11]. H-FIRE delivery results in fewer tissue impedance alterations, creating a more homogeneous treatment field, and subsequently, more uniform ablation zones [17]. Additionally, H-FIRE appears to have greater selectivity for cancer cells, including glioma stem cells, with cancer cell ablation occurring at a lower electric field threshold than that required for ablation of healthy cells [10]. This makes H-FIRE an ideal treatment option for tumors involving critical structures, such as malignant brain tumors, which would otherwise be poor candidates for surgical removal or thermal ablation methods [10].

We have previously demonstrated that H-FIRE, like traditional IRE, induces simultaneous tumor ablation and transient permeation of the BBB. BBBD with H-FIRE is induced with low electric fields and the duration of BBBD is directly related to field strength [18]. When delivered at a voltage to distance ratio of 600 V/cm, BBBD is observed for up to 72 h after treatment in the absence of any concurrent histologic brain tissue damage and cell death. BBB permeability is greatest at 1 h post-H-FIRE treatment, then decreases over time, with BBB recovery completed by 96 h post-treatment [18]. Here, we aim to characterize the mechanisms of H-FIRE-mediated BBBD in vivo using a healthy rodent model. We hypothesize that H-FIRE-mediated BBBD occurs via transient disruption of tight junction complexes of the neurovascular unit secondary to cytoskeletal remodeling and alterations in tight junction protein regulation. Mechanistic characterization of H-FIRE BBBD will be important for translation into brain tumor clinical trials, in which H-FIRE-mediated BBBD will be exploited to deliver therapeutic agents that would otherwise be impermeant to the CNS to target the peritumoral penumbra of the normal-appearing brain.

## 2. Materials and Methods

### 2.1. Assurances and High-Frequency Irreversible Electroporation Delivery

This study was performed in accordance with the principles of Guide for the Care and Use of Laboratory Animals and was approved by the Institutional Animal Care and Use Committee (IACUC #16–156). Forty live adult male Fischer rats weighing 170–215 g were used for this study. The surgical procedures and high-frequency electroporation treatment performed has been described elsewhere [18]. Briefly, rats received a subcutaneous injection of buprenorphine (1 mg/kg, Buprenorphine SR-LAB; Zoopharm, Windsor, CO, USA) prior to anesthesia induction via isoflurane. Anesthesia was maintained with isoflurane (2–3.5%:95% isoflurane: oxygen mixture) delivered via a nosecone. Anesthetized rats were placed in ventral recumbency and positioned in a small animal stereotactic headframe (Model 1350 M; David Kopf Instruments, Tujunga, CA, USA). A unilateral rostrotentorial surgical approach was taken to create a rectangular parietal craniectomy defect in the skull of each rat using a high-speed electric drill (Dremel 3000 Series; Mount Prospect, IL, USA). Following completion of the craniectomy, H-FIRE treatment (voltage-to-distance ratio of 600 V/cm, energized time 100 µs, burst scheme 5-5-5 µs, and 200 bursts) was delivered through two blunt-tipped stainless steel electrodes placed into the brain according to stereotactic coordinates via the micromanipulator arm of the stereotactic frame. Following H-FIRE delivery, craniectomy defects and skin incisions were closed as previously described [18]. Four rats served as sham controls in which the electrodes were placed into the brain, but H-FIRE was not applied. Rats were recovered from anesthesia and monitored until their predetermined survival endpoints (*n* = 4 per time point): 1 h, 24 h, 48 h, 72 h and 96 h.

To assess BBB permeability and provide a visual reference for brain-sample selection for analysis, anesthetized rodents received an IP injection of a solution containing 0.1 mmol/kg gadopentetate dimeglumine (Gd; Magnevist; Bayer, Whippany, NJ, USA) and 75 mg/kg of 2.5% Evans blue dye (EBD; Sigma; St. Louis, MO, USA) 1 h prior to sacrifice [18]. Anesthetized rats were then euthanized by IP pentobarbital (0.5 mL) overdose (Fatal Plus, Vortech Pharm, Dearborn, MI, USA) at the predetermined time-points. Following euthanasia, the brains of rodents were removed and either fixed in 10% neutral buffered formalin solution, or RNA later solution, and 0.5 mL of serum was collected by cardiac puncture. Brains were serially sectioned at 1 mm intervals following fixation for 48 h, then formalin fixed brain slices were paraffin embedded (FFPE) individually in tissue cassettes. RNA later-preserved brains were designated for gene expression and immunoblotting studies, and FFPE specimens for H&E histopathology and immunohistochemistry.

### 2.2. Immunoprecipitation and Western Blotting

Transverse brain slice preparations from each treatment group were sectioned perpendicular to the cerebral longitudinal fissure and separated into treated and untreated hemispheres based on the visualization of parenchymal EBD distribution. Brain tissue samples weighing no more than 30 mg were collected from within the treatment region and used for RNA, DNA and protein extraction. The treatment region within each transverse section was identified by the presence of Evan’s blue dye within the brain parenchyma. Protein for the following experiments were extracted from rat brain tissue lysates using the Qiagen AllPrep DNA/RNA/Protein Mini Kit (Germantown, MD, USA) according to the manufacturer’s instructions.

#### 2.2.1. Tight Junction Proteins

A bicinchoninic acid (BCA) assay (BCA-1; Sigma; St. Louis, MO, USA) was used to determine protein concentrations. Proteins (40 µg/lane) were separated by 12% SDS-PAGE gels and transferred onto polyvinylidene difluoride (Hybond P; Amersham/Sigma; St. Louis, MO, USA) membranes. Membranes were blocked in 5% non-fat milk in Tris-buffered saline with 0.1% Tween-20 at room temperature for 2 h. Membranes were incubated at 4 °C for 12 h with a rabbit polyclonal antibody to ZO-1 (1:800; Invitrogen; Thermo Fisher Scientific; Carlsbad, CA, USA), occludin (1:500; Abcam; Cambridge, MA, USA), claudin-5 (1:1000; Abcam; Cambridge, MA, USA) and β-actin (1:1200; Sigma; St. Louis, MO, USA). Secondary HPR-conjugated goat antibodies against rabbit (1:1000; Beyotime Biotechnology; Shanghai, China) were applied for 2 h at room temperature. Quantitative protein analyses were performed using a digital imaging system (Amersham 600; GE Healthcare, Piscataway, NJ, USA) with dedicated chemiluminescent software (Melanie, 2D gel package; Geneva, Switzerland) and normalized to β-actin (loading control).

#### 2.2.2. Ubiquitin

Protein concentrations were determined by a bicinchoninic acid (BCA) protein assay (Pierce), and 400 µg of total protein from each condition was used for immunoprecipitation. Immunoprecipitation was performed using protein A/G Plus-agarose beads following the manufacturer’s recommendations (Santa Cruz Biotechnology; Santa Cruz, CA, USA). Briefly, lysates were immunoprecipitated with commercial antibodies to occludin-1 (20 µg; Invitrogen, Carlsbad, CA, USA) and claudin-5 (20 µg; Invitrogen, Carlsbad, CA, USA) and subsequently precipitated with protein A/G agarose beads at 4 °C overnight. Samples were rinsed three times with RIPA buffer and then boiled in 40 µL of SDS sample buffer for 5 min. Samples were loaded and resolved on SDS-PAGE (4–20% gradient gels; Thermofisher, Carlsbad, CA, USA) and blotted to 0.2% nitrocellulose membranes (Thermofisher, Carlsbad, CA, USA). Membranes were probed with occludin-1 (1:1000), claudin-5 (1:500) and anti-ubiquitin antibodies (1:500; Invitrogen, Carlsbad, CA, USA). Immunoprecipitated supernatants were also analyzed via immunoblotting with anti-ubiquitin linkage specific K48 (1:1000; Abcam; Cambridge, MA, USA) and K63 (1:1000; Abcam; Cambridge, MA, USA) antibodies. Bound primary antibodies were detected with horseradish peroxidase-conjugated sheep anti-rabbit antibodies (1:20,000; Invitrogen, Carlsbad, CA, USA). Membranes were exposed to Supersignal West Pico chemiluminescent substrate (Invitrogen, Carlsbad, CA, USA) and subsequent signal visualization was performed using a gel documentation system, G:Box Chemi HR16 (Syngene; Frederick, MD, USA).

#### 2.2.3. Cytoskeleton

F-actin and G-actin were evaluated using a G-actin/F-actin in vivo assay kit per the manufacturer’s protocol (BK037; Cytoskeleton, Inc.; Denver, CO, USA). Actin quantitation via SDS-PAGE and Western blot analysis was performed on both products. Proteins were separated by 12% SDS-PAGE gels and transferred to a Western blot membrane according to the manufacturer’s instructions. Membranes were blocked in 5% non-fat milk in Tris-buffered saline with 0.1% Tween-20 at room temperature for 30 min. Membranes were incubated at room temperature for 1 h with a mouse monoclonal antibody to actin (1:1000). Secondary HPR-conjugated antibodies against mouse (1:5000–1:20,000) were applied for 30–60 min at room temperature according to the manufacturer’s instructions. Membranes were processed for chemiluminescent detection of actin (43 kDa). The G-actin standard curve was used to quantify the amount of actin present within lysate supernatants (G-actin) and pellets (F-actin).

### 2.3. Gene Expression Profiling and Pathway Analysis

Total RNA was extracted from rat brains following H-FIRE treatment using the Qiagen AllPrep DNA/RNA/Protein Mini Kit and Qiagen RNeasy Mini Kit (Germantown, MD, USA) per the manufacturer’s protocols. Briefly, brain tissue samples weighing ≤ 30 mg were collected from within the treatment region of transverse brain sections as identified by the presence of Evan’s blue dye within the brain parenchyma. Brain tissue samples were manually homogenized in 600 µL of Buffer RLT Plus (10 µL β-mercaptoethanol per 1 mL Buffer RLT) using a Fisherbrand^TM^ RNase-free disposable pellet pestle (Fisher Scientific, Carlsbad, CA, USA). Total RNA was eluted in 30 µL of RNase-free water and quantified using OD_260_ on a NanoDrop 2000 microvolume spectrophotometer (Thermo Scientific, Baltimore, MD, USA). Total RNA (540 ng) was pooled from 2–4 individual rats per each post-treatment time-point for cDNA synthesis, which was completed using the RT^2^ First Strand Kit (Qiagen, Germantown, MD, USA). Gene expression analyses were performed in triplicate using the RT^2^ Profiler PCR Array for Rat Tight Junctions (Qiagen, Germantown, MD, USA) according to the manufacturer’s protocol via an ABI Fast Block 7500 real-time thermocycler (Applied Biosystems, Memphis, TN, USA). Gene expression data were normalized to internal house-keeping genes and analyzed using the manufacturer’s online software, Gene Globe (Qiagen, Germantown, MD, USA) and Qiagen’s Ingenuity Pathways Analysis (IPA). Fold-change in gene expression relative to sham controls was calculated using the mean Ct value from triplicate PCR reactions. IPA gene expression data were evaluated and ranked based on z-score.

Genes exhibiting the greatest differential expression relative to sham controls based on z-scores generated by GeneGlobe and IPA results were further evaluated with TaqMan qRT-PCR for validation. These included *Cldn5*, *Cldn6*, *Cldn11*, *Cldn12* and *Gnai2*. Total RNA was extracted from rat brains following H-FIRE treatment using the Qiagen RNeasy Mini Kit, as described previously (above). cDNA synthesis via the High-Capacity cDNA Reverse Transcription Kit (ThermoFisher, Carlsbad, CA, USA) was completed using total RNA (800 ng) isolated from 2–4 individual rats per each post-treatment time-point (no pooling) per the manufacturer’s protocol. All primers were supplied by ThermoFisher. A ‘master mix’ was prepared for each individual primer and contained the following per planned reaction: 7µL nuclease-free water, 10 µL Taqman Fast Universal PCR Master Mix (ThermoFisher, Carlsbad, CA, USA) and 1 µL of primer. Triplicate PCR reactions were prepared in a 96-well plate by combining 18 µL of previously prepared ‘master mix’ and 2 µL of cDNA in each well. Quantitative real-time PCR was completed using an ABI Fast Block 7500 thermocycler (Applied Biosystems, Memphis, TN, USA). Mean Ct values calculated from triplicate experiments were used to determine fold change in gene expression relative to sham controls. Lactate dehydrogenase A (*Ldha*) was used as the housekeeping gene to maintain consistency given its role as a housekeeping gene in the completed RT^2^ Profiler PCR Arrays.

### 2.4. Immunofluorescent Imaging

Unstained slides prepared from formalin-fixed paraffin-embedded (FFPE) transverse brain sections were deparaffinized prior to staining according to the following protocol: Slides were washed 2 times in xylene for 5 min each, then 3 times in 100% ethanol for 3 min each, 2 times in 95% ethanol for 3 min each and 2 times in 70% ethanol for 3 min each. Slides were washed 2 times in 1X phosphate-buffered saline (PBS) for 10 min.

#### 2.4.1. Tight Junction Proteins

Antigen retrieval was achieved by immersing deparaffinized slide sections into 1X sodium citrate buffer (10 mM, pH 6, 0.05% Tween 20; Sigma, St. Louis, MO, USA) preheated to 95–100 °C. Slides were allowed to cool on a rocker in the citrate buffer for 20 min then washed 3 times in 1X PBS for 5 min each. Sections were blocked in 2% cold-water fish gelatin (Sigma, Inc., St. Louis, MO, USA) in 0.2% Triton for 1 h, then exposed to mouse anti-claudin-5 (1:100), mouse anti-claudin-6 (1:100) or rat anti-zonula occudens-1 (1:100) in blocking buffer for 24 h at 4 °C. Slides were washed 5 times in 1X PBS with 1% Tween 20 for 5 min each then treated with anti-mouse (1:400) or anti-rat (1:500) Alexa Fluor 594 and allowed to incubate in a dark chamber for 1 h at room temperature. Slides were further washed 4 times in 1X PBS. Negative controls were performed by omitting the primary antibody. Slides were mounted in Fluoroshield^TM^ mounting solution with DAPI counterstain (Sigma, St. Louis, MO, USA) and viewed using a Nikon Eclipse N*i*-U fluorescent microscope equipped with a digital Nikon DS-Ri2 camera and NIS Elements BR software, Version 5.21.01 (Nikon Instruments Inc., Nikon, Japan). Images were captured at the same exposure time to compare changes in expression over time.

#### 2.4.2. Cytoskeleton

Next, 1X Phalloidin-iFluor 488 (ab176753) was prepared in 1X PBS containing 1% bovine serum albumin (BSA). Deparaffinized slides were washed in 1X PBS containing 0.1% Triton for 4 min to permeabilize cell membranes. Slides were washed 2 times in 1X PBS for 3–5 min each, then incubated in the prepared 1X Phalloidin-iFluor 488 solution for 90 min at room temperature in a moist, dark chamber. Slides were washed 3 times in 1X PBS for 5 min each then mounted in Fluoroshield^TM^ mounting solution with DAPI counterstain (Sigma; St. Louis, MO, USA). Slides were viewed and images captured, as previously described in Section 2.4.1.

### 2.5. Statistical Analysis

Gene expression data were analyzed using Ingenuity Pathways Analysis (IPA) and the manufacturer’s array software (Qiagen, Germantown, MD, USA). IPA data were ranked and evaluated based on z-score. Comparisons between individual treatment groups and sham controls were conducted using Student’s *t*-test. Multiple comparisons between treatment groups were conducted using a two-way analysis of variance (ANOVA) followed by a Tukey’s post hoc test. The criterion for significance was set at alpha = 0.05. All data are represented as the mean +/− standard error of the mean.

## 3. Results

### 3.1. H-FIRE-Induced BBBD Is Mediated by Cytoskeletal Remodeling

Given the structural and functional role of actin in anchoring tight junction proteins between endothelial cells and the ability of PEFs to induce cytoskeletal remodeling, we hypothesized alterations in cytoskeletal proteins may contribute to H-FIRE-induced BBBD [19]. Quantitative Western blot analysis for filamentous actin (F-actin) and globular actin (G-actin) revealed a relative decrease in F-actin and a concurrent increase in G-actin at 1 h post-treatment compared to sham controls (Figure 1A,B). Likewise, the F:G-actin ratio was significantly decreased at 1 h post-H-FIRE treatment relative to sham controls, which is consistent with decreased microfilament assembly (Figure 1C).

Immunofluorescent staining for F-actin was performed to further quantify temporal changes in F-actin expression and visualize changes in intracellular distribution, which has been observed previously in cells subjected to PEFs [20]. Phalloidin (F-actin) reactivity was decreased at 1 h post-H-FIRE relative to all other treatment groups and sham controls. Additionally, at the 1 h post-treatment time-point, phalloidin staining was peripherally distributed within cells exposed to H-FIRE treatment, whereas a more diffuse staining pattern was observed within stained tissue sections from all other time points (Figure 1D). Collectively, these results suggest H-FIRE-induced actin disruption increases BBB permeability by impacting the overall organization of tight junction complexes [21].

### 3.2. H-FIRE-Mediated BBBD Is a Transient Process Mediated by Decreases in Tight Junction Protein Concentrations

Intracranial H-FIRE resulted in decreased concentrations of tight junction proteins, claudin-5, occludin and zonula occludens-1 (ZO-1), at 1- and 24 h after treatment relative to sham controls (Figure 2A,B). Tight junction protein concentrations gradually increased starting at 48 h, returning to control levels by 96 h post-treatment, which temporally corresponds with in vivo blood–brain barrier recovery in this model [18]. A similar pattern of tight junction protein expression was observed when transverse brain sections were subjected to immunofluorescent staining for claudin-5 and zonula occludens-1 (ZO-1). A decrease in positive staining for both tight junction proteins was observed at 1- and 24 h post-treatment, followed by a gradual increase in staining over time (Figure 3), consistent with blood–brain barrier recovery.

### 3.3. H-FIRE-Induced Transient BBBD Is Mediated by Alterations in Tight Junction Gene Expression

First, we used the Tight Junction RT^2^ Profiler PCR Array (PARN-143Z; Qiagen, Germantown, MD, USA) to obtain a global view of the impact H-FIRE has on tight junction gene expression and their associated pathways (Figure 4A,B and Appendix A). All tight junction genes with significant changes in expression relative to sham controls are represented in the heat maps produced by GeneGlobe (Figure 4A) and IPA (Figure 4B). In Figure 4A, the magnitude of gene expression is presented relative to the overall average expression of the dataset, whereas gene expression is presented relative to sham controls in Figure 4B. Some of the most significantly downregulated genes following intracranial H-FIRE include *Cldn6*, *Cldn14*, *Cldn15*, *Cldn17*, *Cldn18*, *Igsf5*, *Llgl1,* and *Llgl2*. In contrast, the most significantly upregulated genes include *Gnai2*, *Icam1*, *Ybx3*, *Cldn5*, *Cldn11*, *Cldn12*, *Rac1*, and *Gsk3b*. We were particularly interested in characterizing gene expression changes associated with claudin-5, occludin and zonula occludens-1, given the changes in these protein concentrations that were observed (Figure 2A,B). Thus, occludin (*Ocln*) and zonula occludens-1 (*Tjp1*) were included in Figure 2A even though changes in their mRNA expression relative to sham controls lacked statistical significance. Relative to sham controls, intracranial H-FIRE resulted in a significant upregulation of claudin-5 (*Cldn5*) mRNA expression 1 h post-treatment despite a decrease in detectable claudin-5 protein concentration (Figure 4C). By 24 h post-treatment, the magnitude of *Cldn5* mRNA expression was similar to that observed in sham control samples, followed by a gradual increase in gene expression over time. Maximum *Cldn5* mRNA expression was observed between 72 and 96 h post-treatment, which temporally corresponds to the return of detectable claudin-5 protein concentrations (Figure 2A). In contrast, *Ocln* expression remained slightly decreased across all time points relative to sham controls, although this change was not statistically significant. A small increase in *Tjp1* expression was observed across most time points relative to sham controls, but this was not statistically significant. The top five most significantly altered genes were selected for further analysis based on ranked z-scores determined by Gene Globe and Ingenuity Pathway Analysis (IPA; Web Based Software Analysis Platform; Qiagen; Germantown, MD, USA).

Data collected from the Tight Junction RT^2^ Profiler PCR Arrays were analyzed through IPA software to evaluate the impact of intracranial H-FIRE on tight junction pathways. Results revealed a pattern in which the number and intensity of gene transcription changes increased over time, peaking at 72 h post-treatment. This expression pattern was most compelling in genes associated with claudins. Gene expression changes observed at 96 h post-H-FIRE treatment most resembled those observed at 1 h and 24 h but differed from those observed at 48 h and 72 h. Changes in pathways related to claudins were most striking, with regulation centered around claudin-6 (*Cldn6*) at all time points, which was significantly downregulated relative to sham controls (Figure 5).

*Cldn5* (claudin-5), *Cldn6* (claudin-6), *Cldn11* (claudin-11), *Cldn12* (claudin-12) and *Gnai2* (guanine nucleotide-binding protein Gαi2) exhibited the greatest dysregulation when evaluated using IPA and Gene Globe software platforms, so mRNA expression was individually assessed using qRT-PCR to further validate our initial results. Despite differences in the magnitude of mRNA expression, the temporal pattern of expression was similar across all genes. For all genes, mRNA expression decreased between 1 and 24 h post-treatment, and this difference was significant for *Cldn12*. Similar gene-expression patterns were observed for *Cldn5* and *Gnai2*, characterized by an increase in gene expression between 24 and 48 h post-H-FIRE, a decrease in gene expression between 48 and 72 h post-H-FIRE and an increase in gene expression between 72 and 96 h post-H-FIRE. Significant changes in *Cldn5* expression were observed between 1 and 72 h, and 48 and 72 h post-H-FIRE, with significantly decreased mRNA expression observed 72 h post-H-FIRE treatment. This is in contrast with the pattern of *Cldn5* gene expression observed using GeneGlobe, where expression gradually increased from 24–96 h post-H-FIRE, reaching significance at the 96 h time point. The magnitude of expression changes observed for *Gnai2* failed to reach significance. *Cldn6* expression was significantly decreased relative to sham controls from 24–72 h post-H-FIRE, with significant differences in expression observed between 1 and 48 h, and 48 and 96 h post-H-FIRE treatment. *Cldn11* expression was significantly decreased at 24 h post-H-FIRE, whereas significant increases in *Cldn12* expression were observed at 1 and 72 h post-treatment (Figure 6 and Appendix A).

### 3.4. H-FIRE-Induced BBBD Is Mediated by Post-Translational Modifications to Tight Junction Proteins

Given the relative increase in *Cldn5* mRNA expression despite the relatively decreased CLDN5 protein concentration observed following intracranial H-FIRE treatment, we hypothesized there may be post-translational modifications to account for these discordant results. Co-immunoprecipitation and Western blotting for ubiquitin, occludin, and claudin-5 revealed evidence of increased TJP ubiquitination for up to 48 h after intracranial H-FIRE treatment relative to sham controls (Figure 7A). Immunoblotting for ubiquitin linkage specific K48 and K63 revealed an increase in both K63 and K48 linkage specific ubiquitination of claudin-5 and occludin following intracranial H-FIRE treatment (Figure 7B). Thus, decreased TJP concentrations observed following H-FIRE treatment is predominantly due to TJP protein ubiquitination, which results in both increased proteasomal degradation and endosomal trafficking, although alteration in TJP conformation likely contributes to loss of TJ integrity as well.

## 4. Discussion

Our research team has demonstrated that H-FIRE is capable of transiently permeating the BBB for up to 72 h and successfully characterized the volume of BBBD over time [18]. The current study expands our understanding of H-FIRE induced BBBD by characterizing the molecular mechanisms by which H-FIRE permeates the BBB. Here, we show that intracranial H-FIRE induces transient alterations in tight junction gene and protein expression, while increasing tight junction protein degradation, disrupting tight junction integrity and increasing BBB permeability via paracellular pathways. Additionally, qualitative and quantitative changes in cytoskeletal proteins were observed following treatment with intracranial H-FIRE, which provides additional support for the role cytoskeletal remodeling may have in transiently permeating the BBB.

PEFs, including H-FIRE, are capable of inducing profound cytoskeletal disruption over a range of pulse parameters, and preliminary studies suggest the mechanism of disruption is influenced by pulse length and magnitude [20]. Filamentous structures comprising the cytoskeleton interact with and support the plasma cell membrane through linker proteins that enable mechanical interactions between adjacent cells [20]. Within the BBB, these transmembrane protein interactions are particularly important in establishing tight junction integrity between brain endothelial cells. Zonula occludens proteins, ZO-1, ZO-2 and ZO-3, are intracellular proteins that interact with claudins, junctional adhesion molecules (JAMs) and occludin to form a tight junction protein complex that becomes anchored to the actin cytoskeleton [22,23]. Loss of filamentous actin (F-actin) and redistribution to the cell periphery have frequently been observed following PEFs and persist beyond cell membrane resealing [24,25,26]. The exact mechanisms by which PEF induces cytoskeletal disruption remain to be completely characterized. To date, evidence suggests cytoskeletal disruption likely occurs following direct interactions between the electric field and cytoskeletal proteins, as well as indirect mechanisms linked to disruption of calcium and ATP homeostasis [20,27,28,29,30]. Our results provide additional evidence that H-FIRE is capable of inducing changes in cytoskeletal protein composition by disrupting microfilament assembly. Given the direct interaction between actin and tight junction protein complexes, H-FIRE-mediated actin depolymerization likely plays a critical role in increasing BBB permeability by altering TJP conformation and stimulating endocytosis of TJ components [19,31]. Cytoskeletal remodeling can also lead to increased BBB permeability via redistribution of tight junction complexes within the cell membrane, but the decrease in detectable protein levels we observed via immunoblotting and immunofluorescent staining suggests TJP degradation rather than redistribution [21].

Tight junction gene expression and subsequent protein concentrations were most significantly altered 1 h after intracranial H-FIRE delivery, which temporally corresponds with the maximum volume of BBBD based on our previous work [18]. Despite a decrease in detectable tight junction protein concentrations at 1 h post-treatment, corresponding tight junction mRNA expression was significantly increased. Given the increase in claudin-5 and occludin ubiquitination observed from 1–48 h after treatment, we show that H-FIRE induces increased tight junction protein ubiquitination. Ubiquitination is a type of post-translational modification responsible for regulating protein stability and interactions with other proteins, receptor internalization, as well as enzyme activity. It plays a critical role in TJP recycling and remodeling by signaling endocytosis and possible degradation of intercellular TJPs, which allows for rapid modulation of paracellular BBB permeability in response to physiologic stimuli [32,33]. The fate of ubiquitinated proteins is determined by the number of bound ubiquitin molecules (mono- or polyubiquitination) and the specific lysine (K) residue to which they are bound [34,35]. Polyubiquitination of K48 typically results in proteasomal degradation, whereas K63 linkage specific ubiquitination has multiple functions, including a role in signal transduction and endocytosis [36]. The increase in both K63 and K48 linkage specific ubiquitination of claudin-5 and occludin observed following H-FIRE treatment suggests these TJPs undergo increased proteasomal degradation (K48) and endosomal trafficking (K63). H-FIRE-induced alterations in TJP trafficking ultimately result in decreased TJP concentrations within the endothelial cell membrane, which significantly disrupts tight junction integrity and stimulates tight junction gene transcription in response [37,38]. The end result is an overall decrease in detectable tight junction protein concentrations and increased BBB permeability. The gradual increase in tight junction gene expression observed starting at 48 h post-treatment temporally corresponds with the gradual increase in tight junction protein levels and blood–brain barrier recovery.

We used the results of our initial gene expression and pathway analysis to direct our attention to specific genes which were most significantly impacted by intracranial H-FIRE. As expected, pathways associated with claudin proteins appeared to be most impacted by our treatment given their critical role in tight junction structure and function [23,37,39,40,41]. Claudin-6 (*Cldn6*) is of particular interest, given that changes in gene expression and pathway regulation appeared to be centered around this specific claudin protein. Prior studies suggest claudin-6 is involved in initiating cell adhesion signaling through its role in regulating nuclear receptor gene expression and subsequent activity [42,43]. Expression is predominantly observed in several types of embryonic epithelial cells compared to adult cells. Thus, claudin-6 is highly expressed in a number of germ cell tumors and appears to play an important role in regulating cancer progression [44]. Furthermore, claudin-6 has been extensively studied as a potential prognostic marker and therapeutic target [45,46]. Within the CNS, significantly higher levels of claudin-6 have been documented within choroid tissue compared to BBB-forming microvessels and cortical tissue, and expression decreases with age. Thus, claudin-6 plays an important role in modulating blood-*CSF* permeability, particularly during early development [47]. The specific role of claudin-6 in regulating *blood–brain barrier* function remains to be characterized. PEFs appear capable of inducing neural stem-cell differentiation, which may explain the significance of claudin-6 changes observed within our dataset [48]. Additionally, we cannot completely exclude the presence of varying concentrations of choroidal tissue within our analyzed samples given the proximity of the region of BBBD observed in the rats to the lateral ventricles. Lastly, H-FIRE treatment may result in significant changes to a promoter residing upstream from claudin-6 and indirectly impact its expression.

Of the remaining genes significantly impacted by our treatment, claudin-5 is considered the most abundant and vital to maintaining blood–brain barrier function and is often regarded as the “gatekeeper” of neurological function [49]. Likewise, claudin-11 and -12, along with claudin-1, and -3, also play integral roles in maintaining blood–brain barrier integrity by limiting paracellular diffusion of ions and solutes into the CNS, thereby determining tight junction permeability [49]. Our results suggest intracranial H-FIRE induces alterations in regulation of claudin-5, -11 and -12 at a transcriptional and post-translational level, which are likely imperative to transiently permeating the blood–brain barrier for drug delivery into the CNS. Claudin-17 expression, albeit marginal, has also been demonstrated within the brain, but the role of claudin-17 in maintaining BBB integrity has yet to be described [50]. Claudins-14, -15 and -18 are predominantly expressed outside of the CNS, and do not appear to play a significant role in maintaining BBB function [41,51,52]. Claudin-18 interacts with zonula occludens-2 (ZO-2) in bone to modulate RANKL within osteoclasts; thus, changes in *Cldn18* expression may contribute to BBB permeability. However, the interaction between Claudin-18 and ZO-2 has not been demonstrated within the BBB [53]. The function of GNAI2 in regulating tight junction integrity through its interactions with claudin-5 has previously been demonstrated in human brain endothelial cells [54]. Given the impact of H-FIRE on claudin-5, we would expect to observe significant changes in genes encoding other functionally associated proteins, such as guanine nucleotide-binding protein G(i) subunit alpha-2 (*Gnai2*), and that these changes may parallel one another over time. The discrepancy in *Cldn5* gene expression observed between PCR methods may reflect true differences due to sample preparation, as RNA samples were pooled for analysis using RT2 Profiler PCR Arrays, whereas TaqMan qRT-PCR was performed on individual RNA samples. Additionally, GeneGlobe and IPA platforms use multiple housekeeping genes when performing data analysis, whereas a single housekeeping gene (*Ldha*) was used for statistical analysis of the standard TaqMan qRT-PCR data.

Aside from some claudins, other tight junction genes significantly impacted by intracranial H-FIRE include immunoglobulin superfamily member 5 (*Igsf5*) and LLGL Scribble Cell Polarity Complex Component 1 and 2 (*Llgl1*/2). IGSF5 is predominantly expressed in cells at focal adhesion sites where actin connects to the extracellular matrix, therefore functioning as a cell adhesion molecule [55]. LLGL1/2 are cytoskeletal proteins involved epithelial cell polarity, organization and integrity. Since H-FIRE-induced BBBD occurs secondary to cytoskeletal remodeling, we might expect to observe significant changes in expression of genes with functions related to the cytoskeleton [56,57]. Intracellular Adhesion Molecule 1 (*Icam1*), Y-box Binding Protein 3 (*Ybx3*), Ras-related C3 Botulinum Toxin Substrate 1 (*Rac1*) and Glycogen Synthase Kinase-3 beta (*Gsk3b*) gene expression was significantly increased following intracranial H-FIRE. YBX3 functions as a DNA/RNA binding protein that regulates transcription, translation and DNA repair; thus, the increase in gene expression observed is not surprising given the impact intracranial H-FIRE has on gene regulation [58]. ICAM1 is expressed on brain endothelial cells and plays a major role in immune cell trafficking; thus, H-FIRE appears capable of facilitating local immune cell infiltration [59]. Given the role of RAC1 in regulating BBB stability via modulation of endothelial cell adhesion, cell transformation and inflammation, we would expect gene expression to increase in response to BBB injury in order to promote BBB recovery [60,61]. Likewise, GSK3B controls numerous neuronal functions, including cytoskeletal dynamics and endocytosis, which we have shown are involved in mediating H-FIRE-induced BBBD [62].

The presence of an intact BBB has limited the ability to achieve therapeutic concentrations of many potentially effective chemotherapeutic agents within intracranial tumors and their surrounding penumbra. Endothelial cells, such as those comprising the BBB, appear to be particularly susceptible to PEFs, resulting in increased paracellular permeability, and transient permeabilization of the BBB using PEFs has previously been shown to enhance delivery and pharmacologic effects of chemotherapeutics that would otherwise remain impermeant to the brain [63,64]. This is particularly attractive for treatment of microscopic tumor cell infiltrates extending beyond the gross tumor margin, as these cells represent a major source of disease recurrence and subsequent patient death. Thus, H-FIRE may be exploited to facilitate electrochemoablation of these cells as an adjuvant treatment to gross tumor resection in order to prevent tumor recurrence and prolong overall survival.

## 5. Conclusions

H-FIRE transiently permeates the blood–brain barrier via disruption of tight junction complexes mediated by cytoskeletal remodeling and altered tight junction protein regulation. Our results support the use of H-FIRE-mediated BBBD to facilitate delivery of therapeutic agents to CNS malignancies and improve overall treatment success by targeting infiltrative cells beyond the gross tumor margin.

## Figures and Tables

**Figure 1 biomedicines-10-01384-f001:**
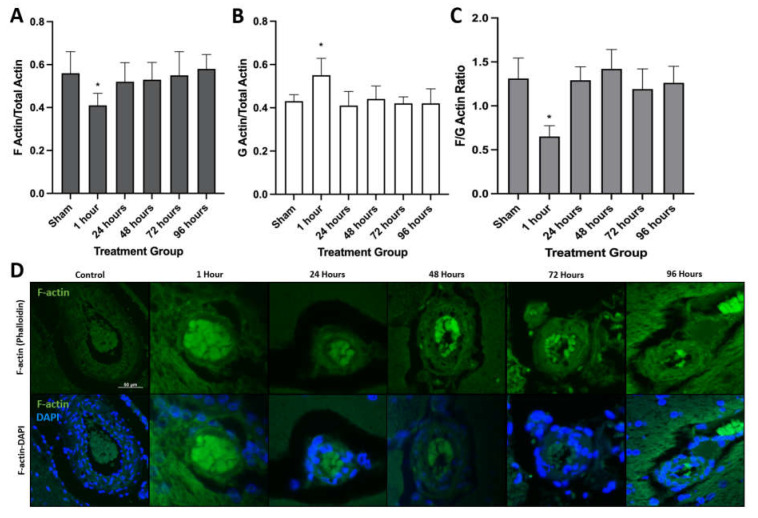
F-actin (**A**) and G-actin (**B**) to total actin ratios demonstrating decreased concentrations of F-actin and increased concentrations of G-actin relative to all other treatment groups. Likewise, the F:G-actin ratio was significantly decreased (* *p* < 0.05) at 1 h post-H-FIRE treatment compared to sham controls (**C**). F-actin (Phalloidin) fluorescence in treated rat brains revealed a decrease in F-actin reactivity with peripheral redistribution at 1 h post-H-FIRE treatment relative to all other treatment groups. All images were captured at 200× magnification (**D**). The scale 50 µm is consistent across all images.

**Figure 2 biomedicines-10-01384-f002:**
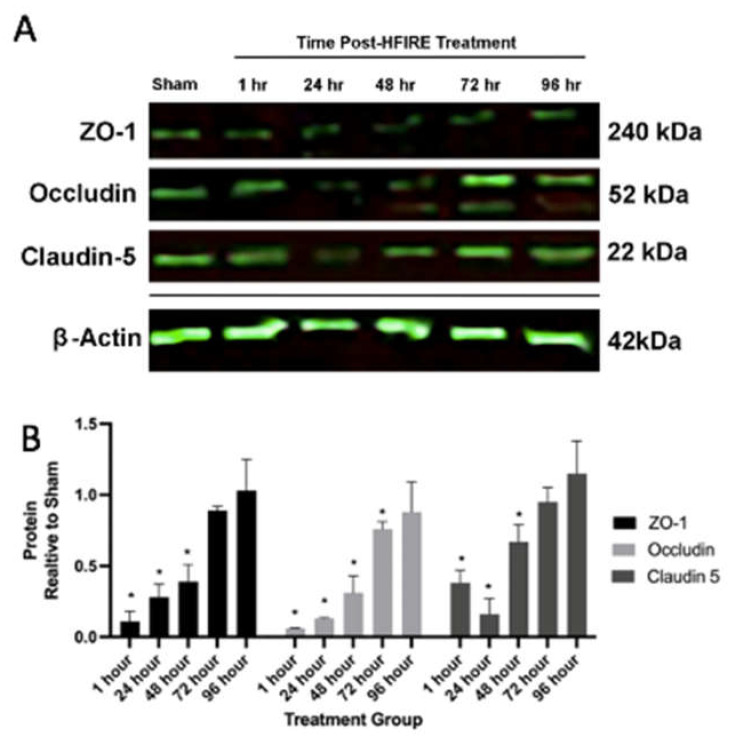
Western blot (**A**) and corresponding protein quantification normalized to β-actin (loading control) (**B**) demonstrating transiently decreased expression of TJPs, zonula occudens-1 (ZO-1), occludin and claudin-5, following H-FIRE treatment relative to sham controls. * indicates statistically significant differences in protein concentrations compared to sham controls (*p <* 0.05).

**Figure 3 biomedicines-10-01384-f003:**
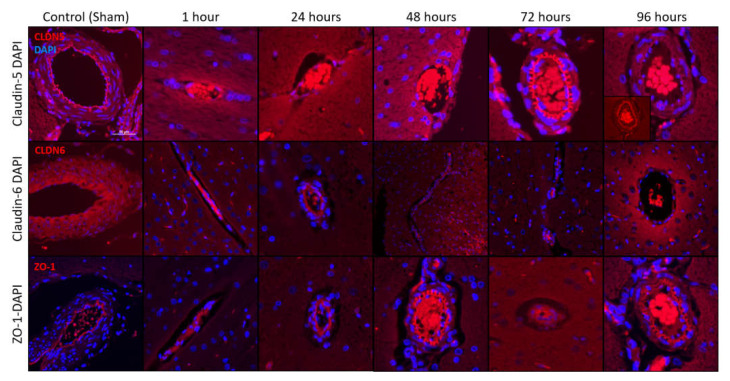
Immunofluorescent staining of transverse brain samples following intracranial H-FIRE revealed decreased claudin-5 (CLDN5) and zonula occludens (ZO-1) reactivity by 1 h post treatment, followed by a gradual increase in reactivity over time. Claudin-6 (CLDN6) reactivity was greatest at 24 h post-H-FIRE treatment, and relatively decreased at all other time points compared to sham controls. Images were captured at 200× magnification. The scale 50 µm is consistent across all images.

**Figure 4 biomedicines-10-01384-f004:**
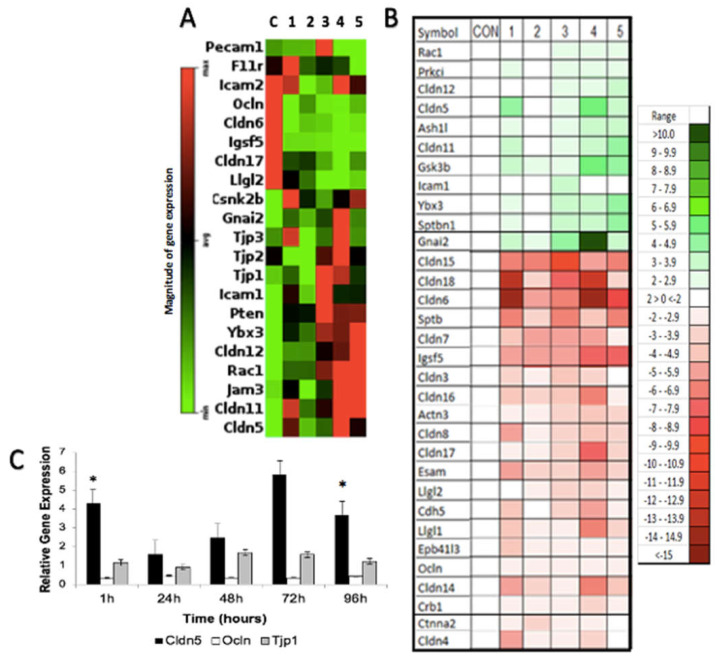
Heat maps of tight junction genes with significant changes in expression (*p* < 0.05) following intracranial H-FIRE generated by GeneGlobe (**A**) (magnitude of gene expression relative to overall average) and Ingenuity Pathway Analysis (IPA) software (**B**) (z-score relative to sham control). C/CON = sham control, Group 1 = 1 h, Group 2 = 24 h, Group 3 = 48 h, Group 4 = 72 h, Group 5 = 96 h post-H-FIRE. *Cldn5* (claudin-5) mRNA expression was significantly increased 1 h after intracranial H-FIRE delivery relative to sham controls, but expression was similar to sham controls by 24 h post-treatment then gradually increased over time. The magnitude of change in *Tjp1* (zonula occludens-1) and *Ocln* (occludin) expression was not significant; however, *Ocln* expression was decreased relative to sham controls at all time points (**C**). * *p* < 0.05.

**Figure 5 biomedicines-10-01384-f005:**
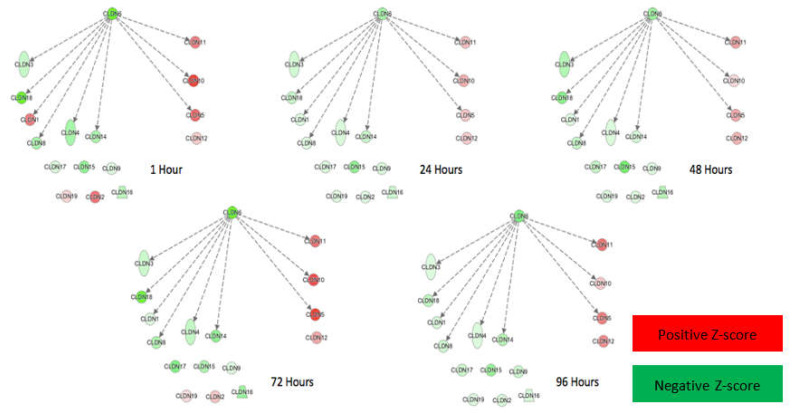
IPA revealed pathways associated with claudins as most impacted by H-FIRE treatment, with changes centered around claudin-6 (*Cldn6*). Genes labeled in red were upregulated and those in green were downregulated relative to sham controls. The intensity of the indicated color corresponds with the z-score magnitude.

**Figure 6 biomedicines-10-01384-f006:**
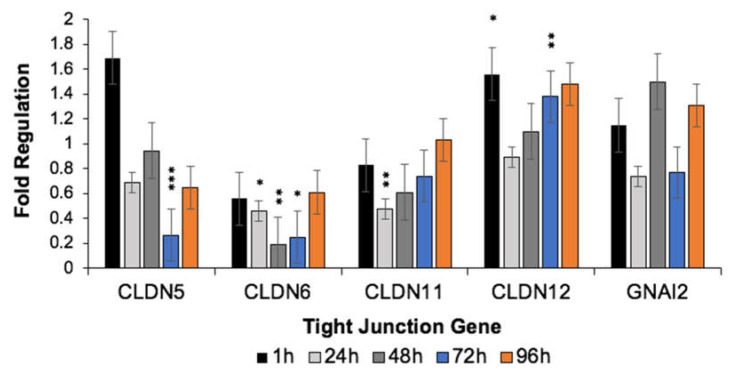
*Cldn5*, *Cldn6*, *Cldn11*, *Cldn12* and *Gnai2* displayed the greatest change in expression based on z-scores generated by IPA and GeneGlobe data analysis, so these genes were further analyzed using RT-PCR. For all genes, mRNA expression decreased between 1 and 24 h post-H-FIRE treatment then increased at a later time-point. *Cldn5* expression was significantly decreased 72 h after treatment compared to the sham control, which is in contrast with results of the RT2 Profiler PCR where *Cldn5* expression was relatively increased at 72 h post-H-FIRE. * *p* < 0.05, ** *p* < 0.01, *** *p* < 0.001.

**Figure 7 biomedicines-10-01384-f007:**
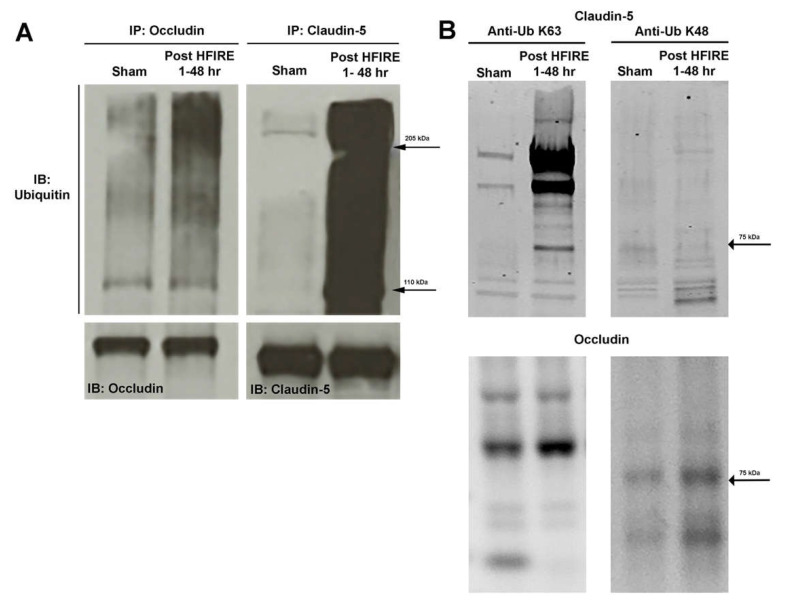
Intracranial H-FIRE resulted in increased occludin and claudin-5 ubiquitination for up to 48 h post-treatment (**A**), indicating post-translational modifications contribute to the inverse correlation observed between levels of mRNA transcripts and protein. Both K63 and K48 linkage specific ubiquitination of claudin-5 and occludin increased after H-FIRE treatment (**B**).

## Data Availability

The data presented in this study are openly available in the Mendeley Data repository at doi: 10.17632/tf2p4wvk29.1, titled “H-FIRE BBBD Mechanism Normal Rat Brain Data”.

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
