# Peer review of "High-Frequency Irreversible Electroporation (H-FIRE) Induced Blood–Brain Barrier Disruption Is Mediated by Cytoskeletal Remodeling and Changes in Tight Junction Protein Regulation"

_biomedicines, 2022, doi:10.3390/biomedicines10061384_

Round 1

Reviewer 1 Report

In the current study, Partridge et al. investigated the molecular mechansims of high-frequency irreversible electroporation induced blood-brain barrier disruption. Disregulation of proteins associated with tight-junction integrity was studied via Western blotting and immunofluorescent staining. While the conclusion is consistent with the results presented, there are a few major issues to be clarified.

Major comments

  • Figure 2A and B: It is unclear whether the protein quantities in the blots were normalized to b-actin signals.
  • Figure 3: Immunofluorescence images didn't always agree with the Western blot results. E.g., Claudin 5 protein amount at 96 h after treatment is comparable to control (Fig. 2B). However, the immunofluorescence image at this time point doesn't show specific claudin 5 staining. Also the ZO-1 staining doesn't fit to Western blot quantification.
  • Fig. 4A and C: According to heat map in 4A, TJP1 is clearly upreagulated (changes from green to red), this doesn't fit to the small changes shown in 4C.

Minor comments:

  • Line 181, 184, and 185: The protein amounts were given as gram, this is obviously not correct.

Reviewer 2 Report

In this manuscript, Partridge and colleagues investigate the molecular mechanisms involved in BBB disruption upon H-FIRE treatment. They observed cytoskeletal remodelling as well as alterations in tight junction proteins expression and post-translation modifications early after treatment followed by complete recovery at 72-96 hours post-treatment. These findings have important implications for the development of H-FIRE as a delivery method for CNS-targeted therapeutics.

In general, I find this manuscript very well written with well-designed experiments. The results support the conclusions made by the authors. I therefore recommend this manuscript for publication following a few minor changes.

Minor modifications:

Line 293: Typo; has been previously been observed

Figure 4: In addition to the five selected genes that demonstrated the most dysregulation in H-FIRE treated animals, the heat map demonstrated several other genes that were significantly reduced or increased by the treatment. Their potential involvement/role in BBB disruption should be discussed in the manuscript.

It is very interesting that the cells would induce increased mRNA expression of tight junction proteins while reducing protein level by ubiquitin-mediated degradation or internalization. It would be interesting to demonstrate recovery of tight junction proteins expression following inhibition of ubiquitination to confirm this is indeed what is happening. Also, perhaps demonstrate what is happening following ubiquitination; degradation in the proteasome or internalization in endosomes for recycling? This could be done through the use of specific inhibitors of those pathways or co-localization studies with known endosomal markers.

Round 2

Reviewer 1 Report

The authors addressed all issued adaquately.